# Formulating Node Labelling as Node Classification or Link Prediction in Different Graph Representations

**Tobias Möller***  
*Institut Polytechnique de Paris*

*tobias@moller.se*

**Borun Shi**  
*Neo4j*

*brian.shi@neo4j.com*

**Reviewed on OpenReview:** *https://openreview.net/forum?id=lK7tjysj0s*

## Abstract

Message-passing Graph Neural Networks (GNNs) are increasingly used for predictive tasks on graphs. Much work has been done to improve GNN architectures, but how the actual data graph should be designed is not well studied. In this paper, we investigate how two different graph representations impact the performance of GNN models across datasets with varying characteristics grouped by homophily, heterogeneity, and number of labels per node. A unique phenomenon is that the same abstract predictive task of labelling nodes is formulated as a node classification problem on one representation and as a link prediction problem on the other. Our work is the first to blur the line between these two basic and fundamental tasks in graph learning. Our experiments on 12 real-world datasets suggest that different representations (and tasks) are optimal for different datasets, models, and hyperparameters. We derive empirical heuristics of choosing between the two and pave the way towards a criterion of choosing the optimal graph representations and towards formally understanding the interconnection between node classification and link prediction.

## 1 Introduction

Message-passing Graph Neural Networks (GNNs)(Hamilton et al., 2018; Veličković et al., 2018; Kipf & Welling, 2017) have been proven to be effective for analyzing graph-structured data and are widely used in numerous domains (Zhou et al., 2021; Hamilton, 2020).

Two of the common tasks on large graphs are node classification (NC) and link prediction (LP). Identifying fraudulant accounts in a financial transaction network(Wang et al., 2023), assigning academic papers with subject categories in a citation graph(McCallum et al., 2000) and classifying types of businesses or products in a Web-based graph(Zeng et al., 2020) are typically defined as NC problems. Recommending friends and users to follow in a social network(El-Kishky et al., 2022) and predicting missing physical or chemical associations in a biomedical graph(Wishart et al., 2017; Szklarczyk et al., 2019) are typical LP problems. These tasks are formulated not only on homogeneous graphs, but also on real-world heterogeneous graphs that have rich node- and edge-type information as well as feature attributes. Several extensions of classical GNN models explicitly leverage heterogeneity(Wang et al., 2019; Chen & Chen, 2021) in an attempt to improve learning performance.

Although GNNs have shown good performance for NC and LP on a variety of graphs, there are also known issues such as over-smoothing(Rusch et al., 2023) and over-squashing(Topping et al., 2022) which limit the expressiveness of GNNs and consequently their performance. The root cause stems from the fact that the computation graph is exactly the same as the data graph, which allows GNNs to possess strong inductive bias. Rewiring methods(Topping et al., 2022; Arnaiz-Rodriguez et al., 2022) that usually involve adding

---

*Work done while the author was at Neo4j.

edges or virtual nodes have been introduced. There is also bag of tricks(Wang et al., 2021; Chen et al., 2022) that incrementally improve GNN performance.

Despite all the progress, very little effort has been put into **how the data graph should be represented** in the first place. Existing work benchmarks on predefined NC and LP tasks on the given representations of datasets. However, there is no canonical way of constructing the data graph and there are numerous ways of representing the same abstract information. Consider the example of modeling users tagging music with genre in a graph, one could define User and Music nodes, and assign Tag edges with properties being a one-hot encoding of genre. An alternative might be having User, Music and Genre nodes, and assigning a User-Music-Genre hyperedge (an edge connecting three nodes). The optimal representation (or schema) depends on the workload, downstream models, and predictive tasks.

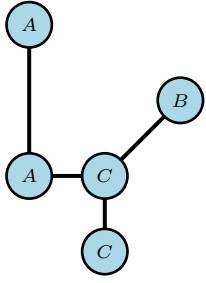
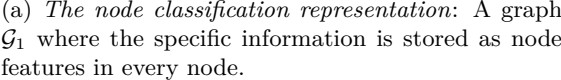
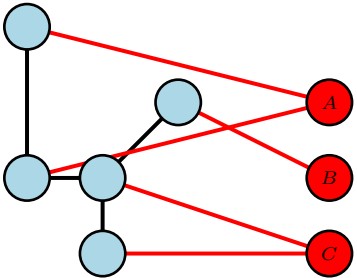

(a) *The node classification representation*: A graph $\mathcal{G}_1$ where the specific information is stored as node features in every node.

(b) *The link prediction representation*: A graph $\mathcal{G}_2$ where the specific information is represented by nodes of different types connected with new edges.

Figure 1: Two ways of representing the same abstract graph $\mathcal{G}$. Colors represent different node and edge types. Letters represent information about nodes.

In this paper, we systematically compare two representations that are general and fundamental, as shown in Figure 1. When we have discrete properties about the nodes, we could represent them as labels on the nodes (Figure 1a), or as distinct nodes with edges connecting them (Figure 1b). This means that for a predictive task on the discrete properties, the task is formulated as NC using the data graph in Figure 1a while it becomes LP using the data graph in Figure 1b. Note that even the formal task differs; a phenomenon of these two representations. This implies that such a decision of picking the data graph early in the machine learning pipeline has profound impact, since entirely separate training and testing processes, such as data splits and metrics, as well as model architectures, can be used and compared across NC and LP for the same end problem. Concretely, we investigate the following research questions:

- How do different representations of the same abstract graph affect the performance of a predictive task defined on it?

- What are the characteristics of a graph that determine whether a node classification representation or a link prediction representation is preferred?

**Contributions.** We give answers to the research questions by evaluating how well different GNNs perform across a wide variety of graphs using NC and LP representations in the transductive setting. We make three key contributions:

- We pose the basic and important question of how the abstract graph should be represented and highlight two general and fundamental representations where even the formal predictive task of NC and LP differs for the same end problem.

- We rigorously map out ways to fairly examine NC and LP representations by defining comparable data splits, evaluation metrics, negative sampling (as a unique step of LP) and model architectures.

- We conduct extensive experiments using datasets of varying characteristics based on homophily index, level of heterogeneity, and number of labels per node.

Our results show that the optimal representation varies for different datasets and depends on the specific GNN used. Although most of the existing benchmarks adopt the NC representation by default, we found that the LP representation indeed sometimes yields better results in fair comparisons. For example, we found that GraphSAGE(Hamilton et al., 2018) is consistently better on the LP version of widely benchmarked ogbn-arxiv(Hu et al., 2020) and similarly GAT(Veličković et al., 2018) on LP versions of DBLP(multi)(Akujuobi et al., 2019) and CiteSeer(Giles et al., 1998). Our work implies that the question of constructing optimal data graphs shall not be ignored and simply using the default representations in benchmarks is insufficient. Our result provides the groundwork for future research towards a formal and general criterion of designing optimal graph representations and a new perspectives which unifies node classification and link prediction.

## 2 Preliminary

**Definition 2.1** (Graph). A graph $\mathcal{G} = (\mathcal{V}, \mathcal{A}, \mathcal{T}, \mathcal{R})$ consists of a set of nodes $\mathcal{V}$, a set of node types $\mathcal{A}$, a set of edges $\mathcal{T} \subseteq \mathcal{V} \times \mathcal{R} \times \mathcal{V}$, and a set of relation types $\mathcal{R}$. Each node $v \in \mathcal{V}$ has one or more node types $\alpha(v) \subseteq \mathcal{A}$. Each edge $(v_i, r, v_j) \in \mathcal{T}$, has a relation type $r \in \mathcal{R}$ which specifies the type of relation between nodes $v_i$ and $v_j$. Each node $v \in \mathcal{V}$ has an initial node embedding $z_v \in \mathbb{R}^d$ for some dimension $d$.

Features for all nodes are abbreviated as $\mathbf{X} \in \mathbb{R}^{|\mathcal{V}| \times d}$. A homogeneous graph has $|\mathcal{A}| = |\mathcal{R}| = 1$. Let $\mathcal{L}$ be a set of labels for the nodes. The partial labelling function $L : \mathcal{V} \to \mathcal{P}(\mathcal{L}) \setminus \emptyset$ assigns some of the nodes their labels. Let $\mathcal{V}_L$ be the domain of $L$. $\mathcal{V}_L \subset \mathcal{V}$ represents the subset of nodes that are labelled. If the graph is not multilabeled, then $|L(v)| = 1$ for all defined $v \in \mathcal{V}_L$.

**Definition 2.2** (Transductive Node Classification). Given a graph $\mathcal{G}$ with node features $\mathbf{X}$ and a partial labelling function $\mathcal{L}$ where $\mathcal{V}_\mathcal{L} \subset \mathcal{V}$, we want to assign labels to the nodes in $\mathcal{V} \setminus \mathcal{V}_\mathcal{L}$ by some $f|_{\mathcal{G}, X, L} : \mathcal{V} \setminus \mathcal{V}_\mathcal{L} \to \mathcal{P}(\mathcal{L})$.

**Definition 2.3** (Transductive Link Prediction). Given a graph $\mathcal{G}$ with node features $\mathbf{X}$ and an incomplete set of edges $\mathcal{T} \subset \mathcal{V} \times \mathcal{R} \times \mathcal{V}$, we want to predict a new set of edges $\mathcal{T}' \subset (\mathcal{V} \times \mathcal{R} \times \mathcal{V}) \setminus \mathcal{T}$ by some $f|_{\mathcal{G}, X} : \mathcal{T}' \to \{0, 1\}$.

**Definition 2.4** (K-hop Homophily Index). The $K$-hop homophily index $\beta_K$ of a graph is given by

$$\beta_K = \frac{1}{|\mathcal{V}|} \sum_{v \in \mathcal{V}} \frac{|u \in \mathcal{N}^K(u) : |L(u) \cap L(v)| > 0|}{|u \in \mathcal{N}^K(u)|}.$$

$\mathcal{N}^K(u)$ is the K-hop neighbourhood of $u$ that includes all nodes that are connected to $u$ with some path of length at most $K$. When $K = 1$ and the graph is single-labelled, our K-hop homophily index reduces to the common homophily index $\beta$ that measures the proportion of nodes that share labels with their neighbours defined in Pei et al. (2020).

### 2.1 Graph Neural Network

A message-passing GNN iteratively aggregates node embeddings from a node's neighbourhood and updates its embedding. These embeddings are initialized as some feature vectors of the nodes. The embeddings and learnable weights are optimised with some objective on the graph. Formally, the message passing step (Gilmer et al., 2017) is

$$\mathbf{h}_v^{(k+1)} = \text{UPDATE}^{(k)}\left(\mathbf{h}_v^{(k)}, \text{AGGREGATE}^{(k)}(\{\mathbf{h}_u^{(k)}, \forall u \in \mathcal{N}(v)\})\right). \tag{1}$$

MEAN, MAX, MIN are commonly used for AGGREGATE. UPDATE is usually some differentiable non-linear function, such as $\text{UPDATE}(x, y) = \sigma(W_1 x + W_2 y)$ with non-linearity $\sigma$ and learnable weights $W_1, W_2$. Note that the neighbourhood function $\mathcal{N}(v)$ does not have to include the full neighborhood of $v$. GraphSAGE(Hamilton et al., 2018) samples the neighbourhoods to improve model scalability. GAT(Veličković et al., 2018) learns attention coefficients between pairs of nodes. AGGREGATE being a set-function ensures the embeddings are

permutation equivariant. GCN(Kipf & Welling, 2016) defines the message passing step as the first-order approximation of localized spectral filters. GIN(Xu et al., 2018) is an architecture that is provably as expressive as the classical WL-test(Weisfeiler & Lehman, 1968) in separating non-isomorphic graphs.

Since the computation structure directly follows the underlying data graph, a strong inductive bias of the underlying graph is baked into the model.

Equation 1 could be applied to heterogeneous graphs $\mathcal{G}$ by simply ignoring the node and edge type information. A natural extension that takes advantage of heterogeneity is to apply Equation 1 separately for each triplet type $\mathcal{A} \times \mathcal{R} \times \mathcal{A}$ as distinct channels, effectively treating them as separate homogeneous subgraphs, and aggregate the messages at the node level. A pictorial illustration of the computation graph is given in Appendix A.1. HAN(Wang et al., 2019) and MAGNN(Fu et al., 2020) use predefined metapaths (list of node and relation types) to induce metapath-induced neighbourhoods as another way to explicitly leverage heterogeneity.

## 3 Node Classification and Link Prediction on Two Representations

Given a graph $\mathcal{G} = (\mathcal{V}, \mathcal{A}, \mathcal{T}, \mathcal{R})$ and the partial labelling function $L : \mathcal{V} \to \mathcal{P}(\mathcal{L})$, we define our representations formally. There are many different representations that can encode the same information specified by $\mathcal{G}$. The same end task may or may not be defined in the same way, depending on the concrete representations. Among many different ways of representing $\mathcal{G}$ for various predictive tasks, we will focus on studying the impact of using the NC representation (Figure 1a) and the LP representation (Figure 1b) for predictive tasks on nodes in the transductive setting.

### 3.1 Description

**Definition 3.1** (Node classification representation). $G_{NC} = (V, A, T, R)$ is our data graph.

**Definition 3.2** (Link prediction representation). $G_{LP} = (V \cup \mathcal{L}, A \cup \{*\}, T \cup T', R \cup \{*'\})$ is our data graph. $\mathcal{L}$ is a new set of nodes disjoint from the original set, $V \cap \mathcal{L} = \emptyset$. $*$ is a new node type where $a(v) = *$ for all $v \in \mathcal{L}$. $*'$ is a new edge type. $T' = \{(v, *', l) \ : v \in V, l \in \mathcal{L}, L(v) \ni l\}$.

With these definitions of concrete graph representations, we can define learning tasks on them. We will define $P_{NC}$ to be the node classification task on $G_{NC}$, and $P_{LP}$ to be the link prediction task on $G_{LP}$.

**Definition 3.3** ($P_{NC}$: Transductive node classification on $G_{NC}$). Given $G_{NC}$ with node features $\mathbf{X}_V$ and partial labelling function $L$, we want to learn $f|_{G_{NC}, X, \mathcal{L}} : V \setminus V_L \to \mathcal{L}$.

**Definition 3.4** ($P_{LP}$: Transductive link prediction on $G_{LP}$). Given $G_{LP}$ with node features $\mathbf{X}_{V \cup \mathcal{L}}$. We want to learn a boolean function $f|_{G_{LP}, X} : (V \times \{*'\} \times \mathcal{L}) \setminus T' \to \{0, 1\}$. The set $\{(v, *', l) : v \in V, l \in \mathcal{L}, f(v, *', l) = 1\}$ therefore represents the predicted edges.

The learned function $f|_{G_{NC}, X, L}$ from task $P_{NC}$ using $G_{NC}$ naturally answers the questions on assigning labels to unlabelled nodes in $\mathcal{G}$. We can interpret the output of $f|_{G_{LP}, X}$ as a function $\tilde{f}|_{G_{LP}, X} : \mathcal{V} \setminus \mathcal{V}_{\mathcal{L}} \to \mathcal{P}(\mathcal{L})$ where $\tilde{f}(v) \ni l \iff f(v, *', l) = 1$. $\tilde{f}|_{G_{LP}, X}$ therefore solves transductive node classification in Definition 2.2. $G_{LP}$ introduces a linear overhead in terms of parameters and time complexity to $G_{NC}$. An example detailed analysis is provided in Appendix A.6, A.7.

### 3.2 Graph Data Splits

Now that we have defined the two tasks $P_{NC}$ and $P_{LP}$ on two graphs $G_{NC}$ and $G_{LP}$, we need to define how splits can be done so that trained models are generalisable (to assign labels for nodes). In order to measure the impact of different representations, we need to ensure that the splits are comparable.

The split for $P_{NC}$ on $G_{NC}$ is simply partitioning the train, validation, and test set where $V_{train} \cup V_{valid} \cup V_{test} = V_{\mathcal{L}}$ where the random partitioning is determined by some seed. While the computation graph provided by $G_{LP}$ is different from that of $G_{NC}$, we need to somehow ensure that the information available

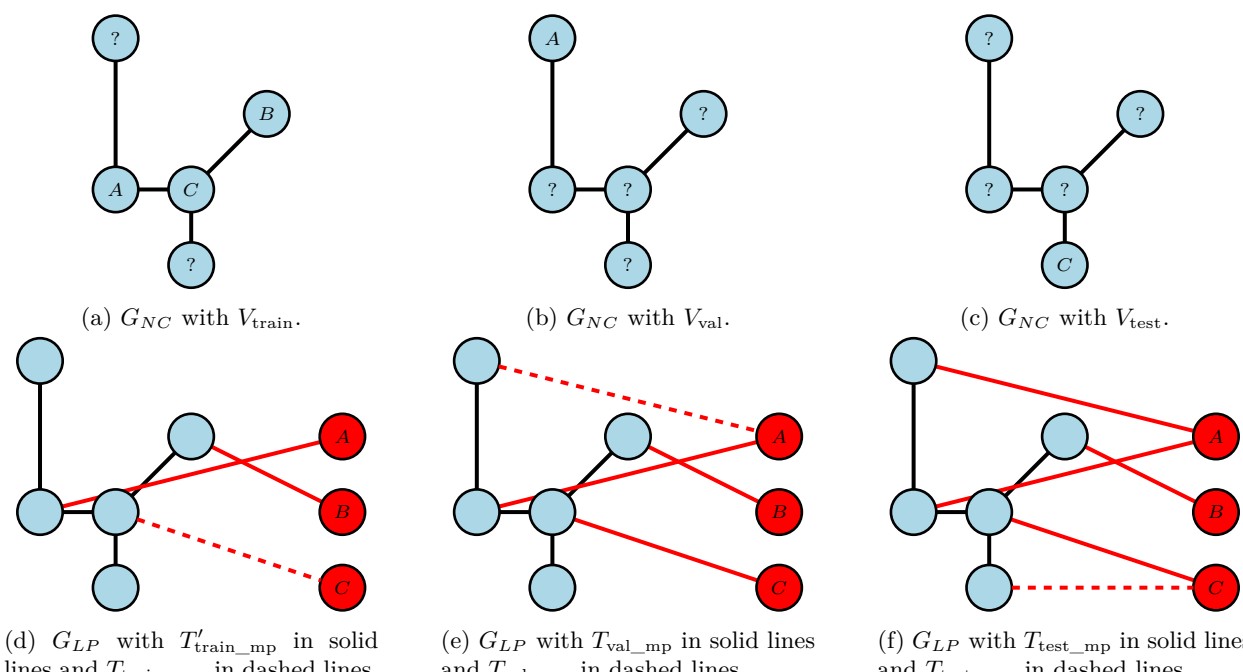

(a) $G_{NC}$ with $V_{\text{train}}$.

(b) $G_{NC}$ with $V_{\text{val}}$.

(c) $G_{NC}$ with $V_{\text{test}}$.

(d) $G_{LP}$ with $T'_{\text{train\_mp}}$ in solid lines and $T_{\text{train\_sup}}$ in dashed lines.

(e) $G_{LP}$ with $T_{\text{val\_mp}}$ in solid lines and $T_{\text{val\_sup}}$ in dashed lines.

(f) $G_{LP}$ with $T_{\text{test\_mp}}$ in solid lines and $T_{\text{test\_sup}}$ in dashed lines.

Figure 2: An example of equivalent splits for $P_{NC}$ on $G_{NC}$ and $P_{LP}$ on $G_{LP}$. The graph $\mathcal{G}$ has 5 nodes and 4 edges with all nodes having some label. (a)(b)(c) show the partition of $V$ into $V_{train}$, $V_{val}$, $V_{test}$ in the $G_{NC}$ representation. (d)(e)(f) show the message-passing and supervision edge splits in $G_{LP}$, created according to Equation 2, 3, 4, 5.

for training the models remains equivalent. This allows comparisons to be made purely on the representations of the information.

The split is less apparent for $G_{LP}$. The set of edges in each split is further divided into message-passing edges (for the computation graph) and supervision edges (for the objective function). Thus, the training data is partitioned and used by the learning method in two different ways, with the partition ratio being a hyperparameter.

Given any $V_{train}, V_{valid}, V_{test}$ for $G_{NC}$, we construct the set of edges for $G_{LP}$ and their message-passing and supervision subsets as:

$$T'_{train} = \{(v, *', l) : L(v) \ni l \wedge v \in V_{train}\}, \ T'_{train\_mp} \cup T'_{train\_sup} = T'_{train}, \ T'_{train\_mp} \cap T'_{train\_sup} = \emptyset \quad (2)$$

$$T_{train\_mp} = T'_{train\_mp} \cup T, \ T_{train\_sup} = T'_{train\_sup} \quad (3)$$

$$T_{valid\_mp} = T_{train\_mp} \cup T_{train\_sup}, \ T_{valid\_sup} = \{(v, *', l) : L(v) \ni l \wedge v \in V_{valid}\} \quad (4)$$

$$T_{test\_mp} = T_{valid\_mp} \cup T_{valid\_sup}, \ T_{test\_sup} = \{(v, *', l) : L(v) \ni l \wedge v \in V_{test}\} \quad (5)$$

$T'_{train\_mp}$ and $T'_{train\_sup}$ in Equation 2 are partitions of $T'_{train}$ according to some supervision ratio hyperparameter. $T_{train\_mp}$ are the message-passing edges, used in the forward pass during training, which means they are used to determine the neighbourhood $\mathcal{N}$ in Equation 1. $T_{train\_sup}$ are the supervision edges, which are used by the objective function (typically cross-entropy) as the positive samples.

Note that as a phenomenon of the equivalence of $P_{LP}$ and $P_{NC}$ for solving the same end task on different graph representations $G_{LP}$, $G_{NC}$, a unique new step of message-passing and supervision data splitting appears since the models solving $P_{LP}$ use information from the graph topology for both the computation graph and the objective, whereas for $P_{NC}$, only the computation graph is based on the topology. An illustration of the equivalent splits is shown in Figure 2.

### 3.3 Negative Sampling

The existence of supervision edges ensures a model generalises beyond seen edges and is able to predict new ones. However, without negative data present in the learning process, the models are likely to converge towards classifying all edges as positive. Since the graph topology is used explicitly as the computation graph for inductive bias in the architecture, as a unique advantage of GNNs, negative samples are normally utilised by the objective function (e.g binary cross-entropy).

There is a rich literature of negative sampling techniques (Nguyen & Fang, 2024; Ying et al., 2018; Yang et al., 2020) for the general link prediction task (of the abstract Definition 2.3) where heuristically "representative and difficult" node pairs are chosen as negative edges. However, for our $P_{LP}$ on $G_{LP}$, we only want to learn a function $f|_{G_{LP},X}$ over $(V \times *' \times \mathcal{L}) \setminus T'$ (i.e between the $V$ nodes to the $\mathcal{L}$ nodes) and not the entire $(V \cup \mathcal{L}) \times (R \cup \{*'\}) \times (V \cup \mathcal{L})$. As a result, we indeed have a true set of negative samples $\{(v, *', l) : v \in V_{\mathcal{L}} \land L(v) \not\ni l\}$. We do not require any negative samples (nor supervision edges) over the subgraph $V \times R \times V$.

## 4 Experiments

We experimentally compare how the two different representations and tasks perform on a variety of 12 real-world datasets and 7 synthetic graphs of different characteristics, using two backbone models (GraphSAGE, GAT) and their heterogeneous extensions as specified in Section 2.1 (Appendix A.1). All experiments use a T4 16G GPU and our code is available on Github[1].

### 4.1 Setup

For each dataset $G$, and for every seed, an 80/10/10 split is obtained on $G_{NC}$ and the equivalent split is created on $G_{LP}$ following Equation 2 3 4 5. 30% of the label edges $T'_{train}$ are encoded as message-passing edges $T'_{train\_mp}$, the remaining 70% of $T'_{train}$ are used as supervision edges $T'_{train\_sup}$. The mutual hyper-parameters (number of epochs and learning rate) and the optimizer (Adam) are the same in both cases. Early stopping is applied on the validation score. The test score with the highest validation score is recorded. Batching is used on larger datasets. For $G_{LP}$, we make sure that all the $\mathcal{L}$ nodes are present in every batch.

Accuracy is calculated for single-labeled datasets, and average precision (AP) is used for multi-labelled ones. We report the average scores for each task/representation and model on each dataset together with the standard deviation. We report a direct comparison of scores for each fixed split and model, on the two representations, and calculate the ratio of LP being better than NC in column $<$. We also report the p-value of the statistics of the NC and LP models, calculated with the independent two-sample unequal-variance t-test (Welch's t-test). Whenever p $< 0.001$, we mark it as 0 in the tables. Small values of p ($< 0.05$) indicate the scores are statistically different, which means we can conclude that the different representations lead to different prediction performances. Additional experimental details are specified in Appendix A.2.

### 4.2 Datasets

We compare three sets of real-world datasets that exhibit different levels of homophily, heterogeneity, and number of labels per node, to identify generalisable trends that correlate to specific dataset characteristics. We additionally use a set of generated random graphs to validate the heuristics we observe.

For homophilic and heterophilic data, six datasets from Rossi et al. (2023) with varying degrees of 1-hop homophily index were used. The homophilic datasets are CiteSeer (Giles et al., 1998; Bojchevski & Günnemann, 2018), Cora-ML (McCallum et al., 2000; Bojchevski & Günnemann, 2018), and ogbn-Arxiv (Hu et al., 2020), and the heterophilic datasets are Chameleon (Rozemberczki et al., 2021), Squirrel (Rozemberczki et al., 2021), and Roman-Empire (Platonov et al., 2024). We additionally provide statistics of K-hop homophily index and neighbourhood sizes for them in Appendix A.3 (Table 9, 10).

---

[1]Github: `https://github.com/tobiasmoller27/nc_lp_paper`

| | GraphSAGE | | | | GAT | | | |
| | NC | LP | $<$ | p | NC | LP | $<$ | p |
|---|---|---|---|---|---|---|---|---|
| CiteSeer | $\mathbf{96.36 \pm 0.90}$ | $94.59 \pm 1.02$ | 0 | 0 | $90.02 \pm 0.94$ | $\mathbf{94.54 \pm 0.85}$ | 100 | 0 |
| Cora-ML | $\mathbf{88.63 \pm 1.18}$ | $86.00 \pm 2.80$ | 0 | 0.018 | $\mathbf{86.60 \pm 1.86}$ | $84.57 \pm 1.86$ | 10 | 0.025 |
| OGBN-Arxiv | $69.53 \pm 0.32$ | $\mathbf{72.17 \pm 0.30}$ | 100 | 0 | $69.76 \pm 0.52$ | $\mathbf{70.40 \pm 0.59}$ | 60 | 0.107 |
| Chameleon | $\mathbf{65.13 \pm 2.60}$ | $57.54 \pm 3.30$ | 0 | 0 | $\mathbf{73.38 \pm 2.18}$ | $67.76 \pm 3.16$ | 0 | 0 |
| Squirrel | $\mathbf{43.88 \pm 1.66}$ | $40.88 \pm 1.09$ | 20 | 0.012 | $\mathbf{44.73 \pm 2.54}$ | $38.15 \pm 2.26$ | 0 | 0.002 |
| Roman Empire | $\mathbf{78.18 \pm 0.69}$ | $76.37 \pm 0.92$ | 0 | 0 | $\mathbf{62.07 \pm 1.09}$ | $56.02 \pm 4.01$ | 10 | 0 |

Table 1: Test scores on the homophilic and heterophilic datasets, using models with two layers. $<$ represents the percentage of runs where $G_{LP}$ is a better representation than $G_{NC}$. p represents the p-value, we report 0 whenever it is smaller than 0.001.

| | GraphSAGE | | | | GAT | | | |
| | NC | LP | $<$ | p | NC | LP | $<$ | p |
|---|---|---|---|---|---|---|---|---|
| CiteSeer | $\mathbf{95.86 \pm 0.85}$ | $95.44 \pm 1.08$ | 50 | 0.347 | $90.12 \pm 1.44$ | $\mathbf{94.35 \pm 0.87}$ | 100 | 0 |
| Cora-ML | $\mathbf{87.37 \pm 1.49}$ | $82.33 \pm 4.73$ | 0 | 0.008 | $\mathbf{85.67 \pm 2.12}$ | $84.70 \pm 2.32$ | 20 | 0.342 |
| OGBN-Arxiv | $70.37 \pm 0.32$ | $\mathbf{72.14 \pm 0.41}$ | 100 | 0 | $\mathbf{69.84 \pm 0.42}$ | $65.59 \pm 1.12$ | 0 | 0 |
| Chameleon | $\mathbf{63.95 \pm 4.07}$ | $56.93 \pm 2.79$ | 0 | 0 | $\mathbf{73.16 \pm 3.28}$ | $59.56 \pm 3.79$ | 0 | 0 |
| Squirrel | $\mathbf{44.12 \pm 2.47}$ | $37.73 \pm 2.59$ | 0 | 0.004 | $\mathbf{42.15 \pm 2.64}$ | $31.96 \pm 4.01$ | 0 | 0.002 |
| Roman Empire | $\mathbf{78.99 \pm 0.65}$ | $74.47 \pm 1.22$ | 0 | 0 | $\mathbf{43.67 \pm 1.79}$ | $37.05 \pm 3.55$ | 10 | 0 |

Table 2: Test scores on the homophilic and heterophilic datasets, using models with three layers. $<$ represents the percentage of runs where $G_{LP}$ is a better representation than $G_{NC}$. p represents the p-value, we report 0 whenever it is smaller than 0.001.

Heterogeneous datasets are ACM, DBLP, and IMDB from Wang et al. (2019) and multi-labelled datasets DBLP-multi (Akujuobi et al., 2019), BlogCat (Tang & Liu, 2009), and Yelp (Zeng et al., 2020).

All of the datasets by default adopt the NC representation and NC benchmarks in the original work. All datasets in $G_{NC}$ form have node features $\mathbf{X}_V$, except BlogCat, which we initialise randomly. For BlogCat, we also found that the split from Tang & Liu (2009) is the only split where the score remained consistent over the training, validation, and test sets, which we use instead of random splits. We initialise $X_{\mathcal{L}}$ for the $\mathcal{L}$ nodes in $G_{LP}$ with one-hot encodings.

### 4.3 Results on Homophilic and Heterophilic Datasets

Tables 1 and 2 present comparisons between NC and LP on homophilic and heterophilic datasets. Both models consistently perform better on the NC representations for heterophilic datasets. This is perhaps counterintuitive since the extra edges $T'$ in $G_{LP}$ directly connect the same-labelled nodes far apart in $G_{NC}$. For homophilic datasets, we observe that GAT on CiteSeer$_{LP}$ performs better than with CiteSeer$_{NC}$. This gap can be as significant as $+4\%$. Furthermore, the advantage of CiteSeer$_{LP}$ is observed consistently across every seed (with LP > NC at 100%).

More interestingly, two-layer GraphSAGE perform better on the LP representations of the widely bench-marked ogbn-arxiv datasets, as opposed to it's default NC presentation. A similar trend is observed with two-layer GAT, although the increase of GAT on ogbn-arxiv$_{LP}$ to ogbn-arxiv$_{NC}$ is not statistically significant ($p = 0.107$). The highest average score is obtained by GraphSAGE on ogbn-arxiv$_{LP}$. The behaviour of three-layer models varies significantly. For example, GAT on ogbn-arxiv$_{NC}$ obtains consistent scores when using two or three layers, but drops significantly on ogbn-arxiv$_{LP}$ with three layers. The behaviour is different for GraphSAGE, where two- and three-layer GraphSAGE consistently and on average performs better on ogbn-arxiv$_{LP}$ than on ogbn-arxiv$_{NC}$.

|  | NC | LP | < | p |
|---|---|---|---|---|
| DBLP (single) | $93.15 \pm 1.13$ | $\mathbf{93.45 \pm 1.61}$ | 50 | 0.636 |
| IMDB | $\mathbf{67.99 \pm 2.26}$ | $56.80 \pm 3.27$ | 0 | 0 |
| ACM | $\mathbf{92.15 \pm 1.71}$ | $36.39 \pm 6.83$ | 0 | 0 |

Table 3: Test scores on the heterogeneous datasets with HAN as the model. $<$ represents the percentage of runs where $G_{LP}$ is a better representation than $G_{NC}$. p represents the p-value, we report 0 whenever it is smaller than 0.001.

|  | GraphSAGE | | | | GAT | | | |
|---|---|---|---|---|---|---|---|---|
|  | NC | LP | < | p | NC | LP | < | p |
| DBLP (multi) | $93.41 \pm 0.55$ | $\mathbf{94.24 \pm 0.26}$ | 100 | 0 | $91.46 \pm 0.41$ | $\mathbf{93.36 \pm 0.32}$ | 100 | 0 |
| BlogCat | $\mathbf{99.83 \pm 0.21}$ | $11.94 \pm 1.20$ | 0 | 0 | $\mathbf{58.08 \pm 0.95}$ | $9.05 \pm 2.07$ | 0 | 0 |
| Yelp | $\mathbf{62.32 \pm 0.55}$ | $39.21 \pm 2.80$ | 0 | 0.003 | $\mathbf{42.59 \pm 0.75}$ | $26.34 \pm 1.09$ | 0 | 0 |

Table 4: Test scores on the multi-labelled datasets, using models with two layers. $<$ represents the percentage of runs where $G_{LP}$ is a better representation than $G_{NC}$. p represents the p-value, we report 0 whenever it is smaller than 0.001.

## 4.4 Results on Heterogeneous Datasets

We experiment with fundamentally heterogeneous datasets and metapath-based heterogeneous GNNs to understand whether the NC and LP representations make a difference in this case. Performance results for heterogeneous datasets using HAN are summarized in Table 3. LP and NC achieve comparable results on DBLP (single). In contrast, NC significantly outperforms LP on IMDB and ACM. The detailed dataset description and the metapaths used are specified in Appendix A.4.

## 4.5 Results on Multi-labeled Datasets

Results in Tables 4 and 5 demonstrate that both two-layer models perform better on DBLP(multi)$_{\text{LP}}$ than on DBLP(multi)$_{\text{NC}}$, and both three-layer models achieve comparable results on the two representations. For BlogCat and Yelp, NC outperforms LP. The extremely large gap shown by BlogCat is not surprising as the dataset is particularly sensitive to splits.

## 4.6 Results on Random Graphs

We plot distributions of the node degrees of the 12 real-world datasets in Figures 5, 6, 7 in Appendix A.3. We observe a pattern in that most of the real-world datasets on which LP performs better (CiteSeer, OGBN-Arxiv, DBLP (multi)) are sparse with many low-degree nodes and few exremely high-degree nodes.

|  | GraphSAGE | | | | GAT | | | |
|---|---|---|---|---|---|---|---|---|
|  | NC | LP | < | p | NC | LP | < | p |
| DBLP (multi) | $93.68 \pm 0.42$ | $\mathbf{93.78 \pm 0.49}$ | 80 | 0.63 | $\mathbf{91.63 \pm 0.49}$ | $91.57 \pm 0.39$ | 60 | 0.766 |
| BlogCat | $\mathbf{99.58 \pm 0.29}$ | $12.59 \pm 1.81$ | 0 | 0 | $\mathbf{34.00 \pm 1.86}$ | $11.36 \pm 1.32$ | 0 | 0 |
| Yelp | $\mathbf{61.41 \pm 0.62}$ | $37.77 \pm 3.02$ | 0 | 0.004 | $\mathbf{40.83 \pm 0.53}$ | $27.05 \pm 1.02$ | 0 | 0 |

Table 5: Test scores on the multi-labelled datasets, using models with three layers. $<$ represents the percentage of runs where $G_{LP}$ is a better representation than $G_{NC}$. p represents the p-value, we report 0 whenever it is smaller than 0.001.

| | GraphSAGE | | | | GAT | | | |
|---|---|---|---|---|---|---|---|---|
| | NC | LP | $<$ | p | NC | LP | $<$ | p |
| bb-100-10-hom | **91.43 ± 5.95** | 90.0 ± 6.88 | 0 | 0.625 | **92.30 ± 5.30** | 90.0 ± 6.19 | 0 | 0.384 |
| bb-100-10-split | **84.29 ± 5.65** | 82.85 ± 5.71 | 10 | 0.578 | 83.81 ± 6.10 | **84.29 ± 5.65** | 10 | 0.858 |
| sbm-5-20-50 | **78.50 ± 8.96** | 74.50 ± 11.5 | 50 | 0.398 | **84.50 ± 10.11** | 75.00 ± 10.72 | 10 | 0.056 |
| sbm-5-1-50 | 75.0 ± 9.21 | **78.0 ± 13.8** | 60 | 0.575 | **81.0 ± 8.31** | 66.0 ± 8.31 | 10 | 0 |
| sbm-1-0.2-1000 | 57.2 ± 2.49 | **89.8 ± 1.14** | 100 | 0 | **89.7 ± 1.23** | 75.9 ± 9.77 | 0 | 0.001 |
| sbm-1-0.2-1000 | 61.4 ± 2.31 | **89.6 ± 1.77** | 100 | 0 | **89.3 ± 1.39** | 81.8 ± 4.29 | 10 | 0 |
| sbm-0.5-0.05-2000 | 60.32 ± 3.81 | **90.69 ± 0.51** | 100 | 0 | **90.51 ± 0.58** | 85.89 ± 6.32 | 10 | 0.046 |

Table 6: Test scores on the random graphs. bb-$m_1$-$m_2$-mode represents a barbell graph of $m_1$ clique sizes and $m_2$ path length with mode either *hom* or *split*. sbm-$p_{same}$-$p_{diff}$-blocksize represents a sbm graph of four blocks of blocksize number of nodes, $p_{same} \times 10^{-2}$ probability of edges in a block and $p_{diff} \times 10^{-2}$ probability of edges across block. When $p_{same} > p_{diff}$ the graphs are generally homophilic. When $p_{same} < p_{diff}$ the graphs are generally heterophilic. Two sbm-1-0.2-1000 graphs are generated with different seeds.

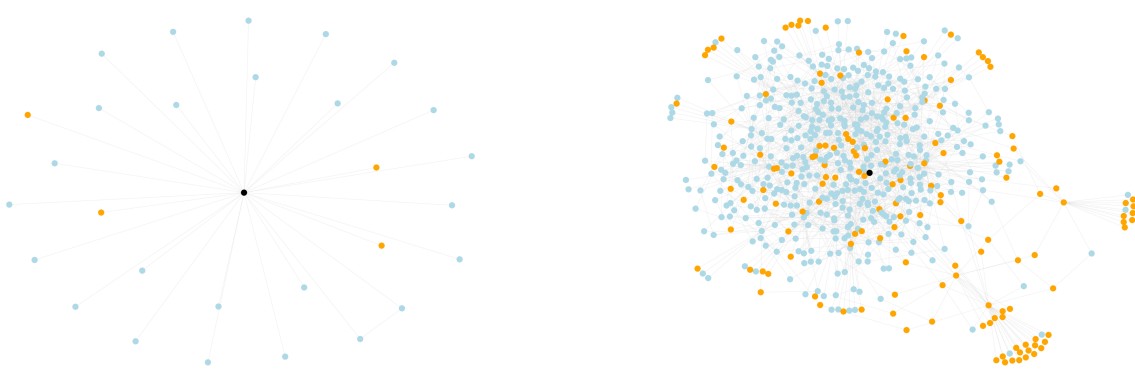

(a) 1-hop ego graph (colored by true label) of node id 720.

(b) 2-hop ego graph (colored by true label) of node id 720.

Figure 3: Example one-hop and two-hop neighbourhoods of a node id 720 in the sbm-0.5-0.05-2000 graph in $G_{NC}$ representation. Node 720 is highlighted in black. Nodes share the same label are in lightblue. Nodes of the other label are in orange.

Heuristically, there seems to be a trend of *LP is sometimes better than NC on sparse and weakly homophilic graphs with few high-degree nodes*. We extend our experiments to a set of random graphs generated under different distributions and various label assignments.

A barbell graph consists of two cliques of $m_1$ number of nodes connected by a path of length $m_2$. We assign binary labels with a method *hom*, where each clique with half of the path being one label. We assign binary labels with another method *split*, where both cliques are assigned the same label, and the path the other label. Both methods produce homophilic graphs. A stochastic block model (sbm) graph consists of k blocks with probabilities $p_{same}$ and $p_{diff}$ of creating edges over nodes of the same and different blocks respectively. We create sbm graphs with four blocks of varying sizes and edge probabilities. We assign binary labels: one label for two blocks of nodes, and another label for the other two blocks. We randomly flip node labels for 10% of nodes for all graphs. While the graph generation and node labelling process is random, the graph function to learn is non-trivial. The label flipping introduces additional noise which the models need to learn to not overfit on. We generate all $G_{NC}$ graphs and transform them into $G_{LP}$.

| | LPA | | | |
|---|---|---|---|---|
| | NC | LP | $<$ | p |
| CiteSeer | $\mathbf{93.71 \pm 1.67}$ | $93.59 \pm 1.56$ | 10 | 0.873 |
| Cora-ML | $\mathbf{87.03 \pm 2.11}$ | $86.87 \pm 2.03$ | 0 | 0.859 |
| OGBN-Arxiv | $\mathbf{73.33 \pm 0.28}$ | $73.27 \pm 0.29$ | 20 | 0.626 |
| Chameleon | $\mathbf{40.13 \pm 0.04}$ | $39.65 \pm 3.99$ | 10 | 0.792 |
| Squirrel | $\mathbf{30.23 \pm 1.59}$ | $29.90 \pm 1.54$ | 10 | 0.647 |
| Roman Empire | $\mathbf{5.56 \pm 0.33}$ | $5.52 \pm 0.39$ | 50 | 0.810 |
| DBLP(multi) | $90.05 \pm 0.41$ | $\mathbf{91.08 \pm 0.41}$ | 100 | 0 |
| BlogCat | $13.43 \pm 0.18$ | $\mathbf{16.97 \pm 0.31}$ | 3 | 0 |
| Yelp | $\mathbf{32.47 \pm 0.18}$ | $32.08 \pm 0.17$ | 0 | 0 |

Table 7: Test scores using LPA with three layers. $<$ represents the percentage of runs where $G_{LP}$ is a better representation than $G_{NC}$. p represents the p-value, we report 0 whenever it is smaller than 0.001. Note that the LP column represents running LPA on a variant of $G_{LP}$ to classify the original set of nodes, therefore not solving the link prediction task.

Our result is shown in Table 6. Similar trends observed in real-world datasets can be found again. For example, NC is better for dense (homophilic) graphs bb-100-10-*. For sparse and weakly homophilic graphs such as sbm-0.5-0.05-2000, LP is +30% better for GraphSAGE than NC on average and for every split. As the graphs become more sparse, the performance of GAT improves on LP but not NC. As an example, we plot in Figure 3 the one-hop and two-hop neighbourhoods of a particular node in sbm-0.5-0.05-2000$_{NC}$ which LP classified correctly but not NC with GraphSAGE. The interpretation is that while the node is homophilic (over both one and two hops), there is still a portion of nodes of the opposite label. However, under the $G_{LP}$ representation, it's two-hop neighbourhood of same-labelled nodes will explode disproportionately. This dominant scaling of neighbourhoods over higher k-hops imply more message passing from same-labelled nodes in $G_{LP}$.

## 4.7   Results on Additional Models

In addition to GraphSAGE and GAT, there are many other alternative models as well as classical algorithms that can be used to label nodes. One such example is the label propagation algorithm (LPA)(Zhu & Ghahramani, 2002). Message-passing GNNs could be viewed as a continuous extension of LPA where the propagation and aggregation functions are parameterised and learnt. LPA is fundamentally a node classification algorithm that cannot solve $P_{LP}$. However, we can still apply LPA on a variant of $G_{LP}$, by keeping all the labels on $V$, and solve the node classification problem on the subset of nodes $V$ (and not the $\mathcal{L}$ nodes). In this setup, the added $\mathcal{L}$ nodes and $T'$ edges according to Definition 3.2 are simply rewiring the graph. We run LPA on the homophilic/heterophilic and multi-labelled datasets and report the results in Table 7.

The performance of (3-layer) LPA closely positively correlate with the level of (3-hop) homophily index (Appendix A.3 Table 9) on both $G_{NC}$ and the rewired $G_{LP}$. Perhaps surprisingly, for all the heterophilic/homophilic datasets, LPA yields comparable results (any difference is statistically insignificant). However, LPA performs better on the rewired $G_{LP}$ graphs for DBLP(multi) and BlogCat. Studying LPA on $G_{NC}$ and $G_{LP}$ will likely offer intermediate insights towards formally and fully bridging $P_{NC}$ and $P_{LP}$.

We also experiment with GCN(Kipf & Welling, 2016) and GIN(Xu et al., 2018) which are common GNN baselines. However, neither models work out-of-the-box on $G_{LP}$ since they are unable to handle heterogeneous graphs. We describe in detail the necessary modifications as well as experimental results for GCN in Appendix B and for GIN in Appendix C. Neither model yields insightful results for $P_{LP}$ likely due to the adhoc heterogeneous adaptation we have to make.

|  | GraphSAGE | | GAT | |
|---|---|---|---|---|
|  | LP | $\downarrow \Delta$ | LP | $\downarrow \Delta$ |
| CiteSeer | $95.34 \pm 0.97$ | $0.1\%$ | $94.06 \pm 1.49$ | $0.29\%$ |
| OGBN-Arxiv | $69.63 \pm 1.52$ | $2.51\%$ | $59.84 \pm 5.23$ | $5.75\%$ |
| Roman Empire | $66.19 \pm 1.86$ | $8.28\%$ | $34.74 \pm 5.65$ | $2.31\%$ |

Table 8: Test scores when using 80% message-passing edges, $|T'_{train\_mp}/T'_{train\_sup}| = 8/2$ instead of 3/7. $\downarrow \Delta$ represents the changes in average scores.

### 4.8 Hyperparameter Analysis

There is a spectrum of model architecture and learning hyperparameter choices that impact the performance of different models to solve $P_{NC}$ on $G_{NC}$ and $P_{LP}$ on $G_{LP}$. A hyperparameter unique to LP but not NC is the training message-passing and supervision edge ratio. Table 8 shows the results when we use 80% edges for message passing instead of 30%. We also examine the impact of embedding dimensions in representative datasets in Appendix A.5 Table 12. Variations for different datasets validate that the common impact of hyperparameter choices carry over to the different graph representations.

## 5 Related Work

**Rewiring.** There are several recent works that modify the computation graphs from the underlying data graph in order to remediate over-smoothing or over-squashing effects of GNNs, or to improve empirical performance. Topping et al. (2022) reduces over-squashing by removing bottlenects in the graph by examining their curvature. Barbero et al. (2024) improves connectivity while preserving locality (distant nodes are kept separate). Qian et al. (2024) introduces virtual-nodes that allows efficient message passing over originally long-distance nodes. Rossi et al. (2023) show that treating heterophilic graphs as directed can improve the performance of GNNs when they are appropriately extended to directed-GNN. The approach of rewiring often introduces nodes or edges that are not part of the data whereas we take the perspective of considering different data representations on an abstract graph. Unique to our two representations, the appropriate formal task becomes different, whereas none of the rewiring methods affect the end task.

**Using label information.** Our LP representation encodes the label information of nodes as new nodes and edges in the concrete data graph itself. There are several other works that utilise node-label information for learning. Wang et al. (2021) combines node label and feature information together by propagating them in parallel. This approach dates back to the classical label propagation algorithm(Zhu & Ghahramani, 2002), with Sato (2024) extending them by introducing traning-free GNNs that approximate label propagation.

**Node classification and link prediction.** There is very limited work that connects these two basic and fundamental learning tasks on graphs. Fadaee & Haeri (2019) are the first to point out a connection between classification and link prediction. They propose an algorithm that labels Euclidean data by using k-nearest neighbour to create a graph and apply a variant of the Adamic-Adar index(Adamic & Adar, 2003) on the constructed graph. Our work could be considered an extension of their work to the context of learning on graphs, in particular as a step towards answering the general question of how representations affect message-passing GNNs. Abboud & İsmail İlkan Ceylan (2021) studies jointly the task of transductive node classification over incomplete graphs and link prediction over graphs with node features. Wu et al. (2022), Bilgic et al. (2007) similarly jointly learns node classification and link prediction. Daza et al. (2021) evaluates entity representations that are learnt with LP objective.

## 6 Conclusion

In this paper, we investigate how different representations of the same abstract graph affect the performance of a predicted task defined on it. We study one of the most common predictive tasks of labelling nodes. It turns out that among many different possible general representations, one could assign nodes with labels

and formulate the task as a node classification problem, or construct new label-nodes, with edge connections representing label assignment to other nodes, thereby formulating the task as a link prediction problem.

Our work makes a connection between two basic and fundamental tasks in graph learning at the input graph representation level. The line between benchmarking on each task is now blurred. Our experiments show that different representations and their respective tasks are optimal for different graphs, models and hyperparameters. The majority of such predictive tasks have been only benchmarked as a node classification problem, which we now know is insufficient. In line with many recent works that challenges graph learning benchmarks(Luo et al., 2024; Lv et al., 2021; Bechler-Speicher et al., 2025; Coupette et al., 2025) on the use of complex GNNs, graph structures themselves and hyperparameter choices, our work adds a new consideration to all past and future benchmarks.

Our results also empirically suggest heuristics of when a representation is better than the other. It appears that relatively homophilic sparse graphs with few high-degree nodes may be suitable for link prediction representations. We also observe that several of the graphs that link prediction perform better on are subsets of citation networks, suggesting correlation with more nuanced characteristics, such as node features or higher-order graph properties.

We pave the way for formally and fundamentally understanding the interconnection between the key problems of node classification and link prediction. We demonstrate the importance of considering different graph representations and call for more comprehensive future benchmarks. We hope our work is a step towards a general and provable criterion of defining the optimal representation and task.

## 7    Limitations and Future Work.

Our paper is empirical in nature. Given that there is no prior work on the question studied, we attempt to first answer the question of whether different representations matter. The answer turns out to be positive. A valuable future direction will be formally establishing the connections between different representations, in particular for the node classification and link prediction representations, as well as the respective tasks.

Our work focuses only on the transductive setting due to the amount of interdependent components in the entire learning process, such as initialisation, model choices, model and training process hyperparameters, and datasets. Future empirical work could extend our problem definitions to inductive cases. Such an extension will be nontrivial, for example, the optimal way of defining and sampling negative edges in the inductive case is unobvious.

### Acknowledgments

This work is an extension of Tobias Möller's master's thesis(Möller, 2025) while at Neo4j and Lund University. The authors thank Alexandros Sopasakis and Donglin Liu (Lund University) for helpful discussions and comments.

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

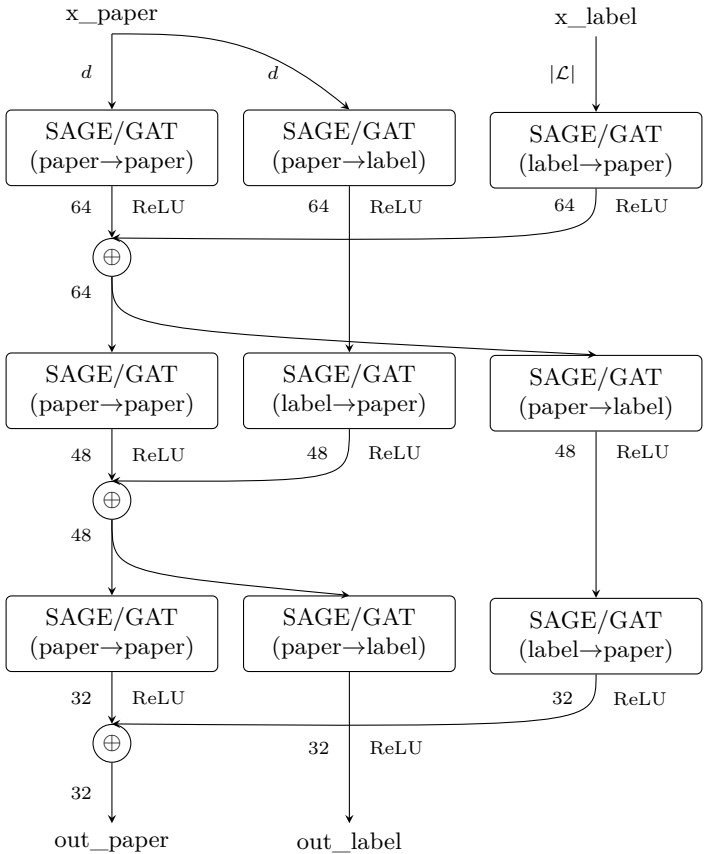

Figure 4: The architecture of a heterogeneous GNN for a graph with two node types $\mathcal{A} = \{\text{paper, label}\}$ and one relation type $\mathcal{R} = \{*\}$ with no edges between label nodes. This example GNN has three layers. A separate channel is instantiated for each triplet type. $\oplus$ denotes an aggregation of embeddings, in this paper, PyG's default aggregator, summation, is used.

## A  Appendix

### A.1  Computation graph of heterogeneous GNN

An example computation graph of a heterogeneous GNN is illustrated in Figure 4. This implementation is directly supported by PyG's *to_hetero()* function on *HeteroData*(Fey & Lenssen, 2019). This implementation is a basic and natural extension of homogeneous GNNs without adding metapaths which represent additional heuristics that strengthen the models. The models with three layers have layer sizes (64, 48, 32), while the models with two layers have layer sizes (64, 32). In the GAT models, four attention heads are used. For both task representations, the decoder is a single linear layer on either nodes (for NC) or node-pairs (for LP).

### A.2  Additional experimental setups

For the smaller real-world datasets and all synthetic graphs we used 10 random seeds. For larger datasets, use 5 seeds for obgn-arxig and Squirrel and 3 seeds for BlogCat and Yelp if the results are consistent. Default learning rate = 0.01. Default hidden channels = 64 and embedding size = 32.

The homogeneous GraphSAGE and GAT encoders consist of either two or three SAGE / GAT-layers with RELU-activation functions between them. The models with three layers have layer sizes (64, 48, 32), while the models with two layers have layer sizes (64, 32). In the GAT models, four attention heads are used. For

|  |  | 1-hop | 2-hop | 3-hop | RW 2-hop | RW 3-hop |
|---|---|---|---|---|---|---|
| Homophilic | CITESEER | 0.96 | 0.94 | 0.92 | 0.95 | 0.93 |
|  | CORA-ML | 0.81 | 0.73 | 0.58 | 0.77 | 0.61 |
|  | OGBN-ARXIV | 0.64 | 0.50 | 0.37 | 0.58 | 0.42 |
| Heterophilic | CHAMELEON | 0.25 | 0.25 | 0.23 | 0.30 | 0.23 |
|  | SQUIRREL | 0.22 | 0.21 | 0.20 | 0.26 | 0.21 |
|  | ROMAN-EMPIRE | 0.05 | 0.07 | 0.07 | 0.29 | 0.11 |

Table 9: Homophily numbers for the different datasets in the heterophilic and homophilic comparison, before and after rewiring with an 80/10/10 split. RW denotes homophily numbers for the rewired graph.

|  | 1-hop | 2-hop | 3-hop | RW 1-hop | RW 2-hop | RW 3-hop |
|---|---|---|---|---|---|---|
| CITESEER | 2.5 | 16.7 | 42.5 | 2.76 | 57.6 | 186.9 |
| CORA-ML | 5.4 | 71.8 | 294.7 | 5.7 | 99.3 | 422.3 |
| OGBN-ARXIV | 13.7 | 3484.2 | 18469.6 | 13.9 | 4151.7 | 22579.6 |
| CHAMELEON | 27.6 | 558.7 | 1066.3 | 27.8 | 580.8 | 1312.3 |
| SQUIRREL | 76.3 | 1692.1 | 3538.5 | 76.5 | 1736.1 | 3836.9 |
| ROMAN-EMPIRE | 2.9 | 7.1 | 12.8 | 3.1 | 121.2 | 750.6 |

Table 10: Neighbourhood sizes for the different datasets in the heterophilic and homophilic comparison, before and after rewiring with an 80/10/10 split. RW denotes neighborhood sizes for the rewired graph.

both task representations, the decoder is a single linear layer on either nodes (for NC) or node-pairs (for LP).

The HAN model only consists of one layer, since the metapaths allow for message passing in neighbourhoods larger than one-hop. Following Wang et al., eight attention heads are used together with a dropout rate of 0.6, and 128 hidden channels. The decoder architectures are the same as for GraphSAGE and GAT.

## A.3 Dataset Statistics

Number of nodes, number of edges, number of classes, average degree, and node feature size for every dataset can be found in Table 11. For each dataset, the node degree distribution is visualised in Figures 5, 6, and 7. For the heterogeneous datasets, the node degree distribution is calculated on the computation graph induced by the meta-paths.

## A.4 Heterogeneous Dataset Information

ACM is a network of papers, authors, and subjects, where the papers are divided into three classes (*database*, *wireless communication*, *data mining*). Paper features correspond to elements of a bag-of-words representation of keywords. The meta-path set {*PAP*, *PSP*} is used in Wang et al. (2019).

DBLP is a network of authors, papers, conferences and terms. The labels indicate research areas of the authors, and are one of *database*, *data mining*, *machine learning*, and *information retrieval*. Author features contain the elements of a bag-of-words represented by keywords. The meta-path set employed in Wang et al. (2019) is {*APA*, *APCPA*, *APTPA*}.

IMDB is a dataset containing movies, actors, and directors. Each movie is divided into one out of three classes (*Action*, *Comedy*, *Drama*). Movie features correspond to elements of a bag-of-words representation of plots. The meta-path set {*MAM*, *MDM*} is used in Wang et al. (2019).

The new $G_{LP}$ representations of these graphs make it possible to define new meta-paths which include label nodes. Studying this is out of the scope of this paper. The same meta-path sets are used as in the

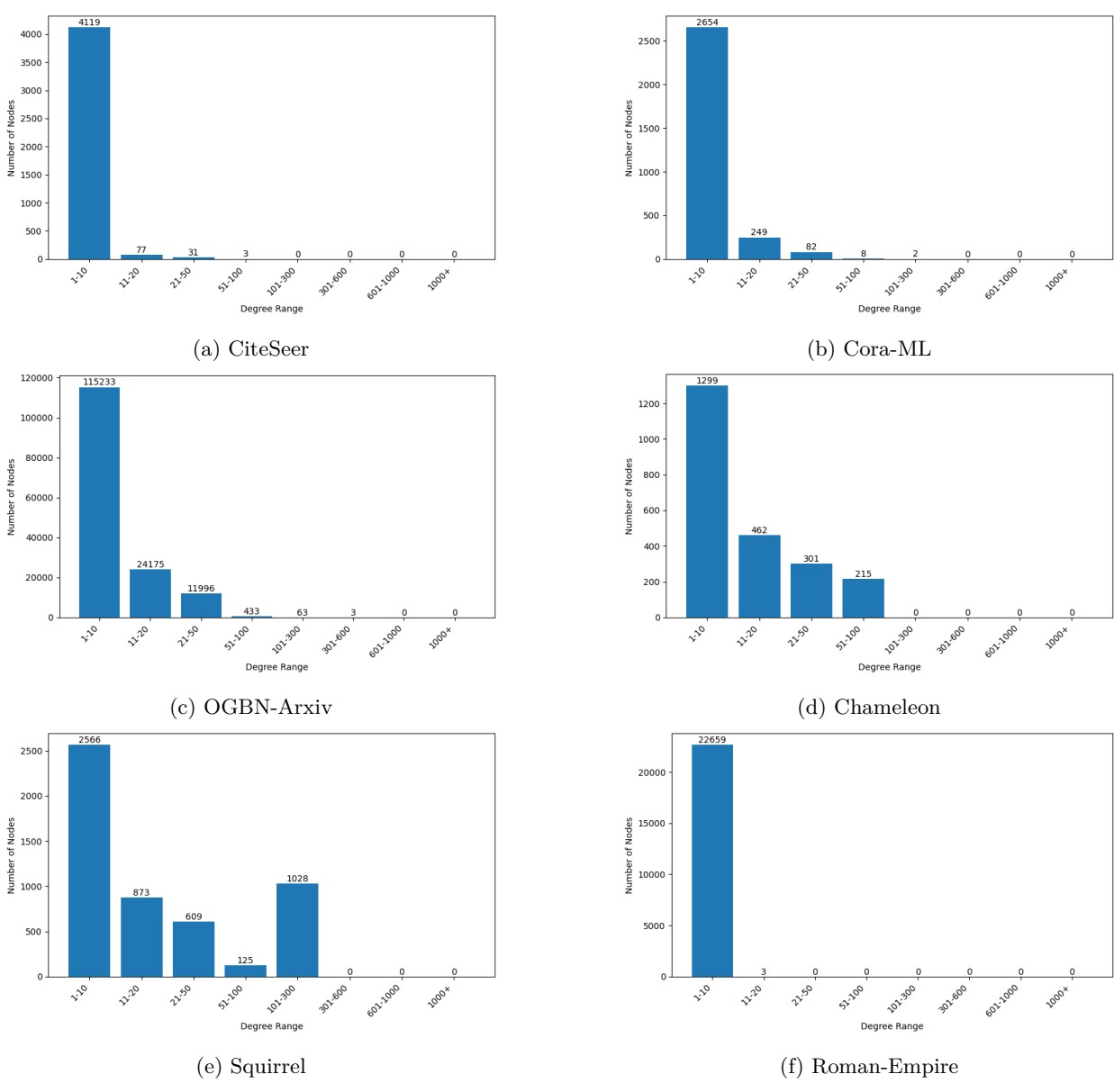

Figure 5: Node degree distributions for the six datasets with varying label homophily levels.

|  | $|V|$ | $|T|$ | Number of Classes | $d$ | Average Degree |
|---|---|---|---|---|---|
| CiteSeer | 4,230 | 10,674 | 6 | 602 | 2.5 |
| Cora-ML | 2,995 | 16,316 | 7 | 2,879 | 5.4 |
| OGBN-Arxiv | 169,343 | 1,166,243 | 40 | 128 | 13.7 |
| Chameleon | 2,277 | 36,101 | 5 | 2,325 | 27.6 |
| Squirrel | 5,201 | 217,073 | 5 | 2,089 | 76.3 |
| Roman Empire | 22,662 | 65,854 | 18 | 300 | 2.9 |
| DBLP (single) | 4,057 | 119,783 | 4 | 334 | 5,942.9 |
| IMDB | 4,932 | 20,172 | 2 | 3,489 | 48.1 |
| ACM | 3,025 | 273,000 | 3 | 1,902 | 4,124.3 |
| DBLP (multi) | 28,702 | 68,335 | 4 | 223 | 4.8 |
| BlogCat | 10,312 | 333,983 | 39 | 0 | 64.8 |
| Yelp | 716,847 | 7,340,000 | 100 | 300 | 19.5 |

Table 11: Statistics of all real-world datasets used in the paper, where $|V|$ is the number of nodes, $|T|$ is the number of edges, and $d$ is the size of node features. All statistics are based on the $G_{NC}$ representation.

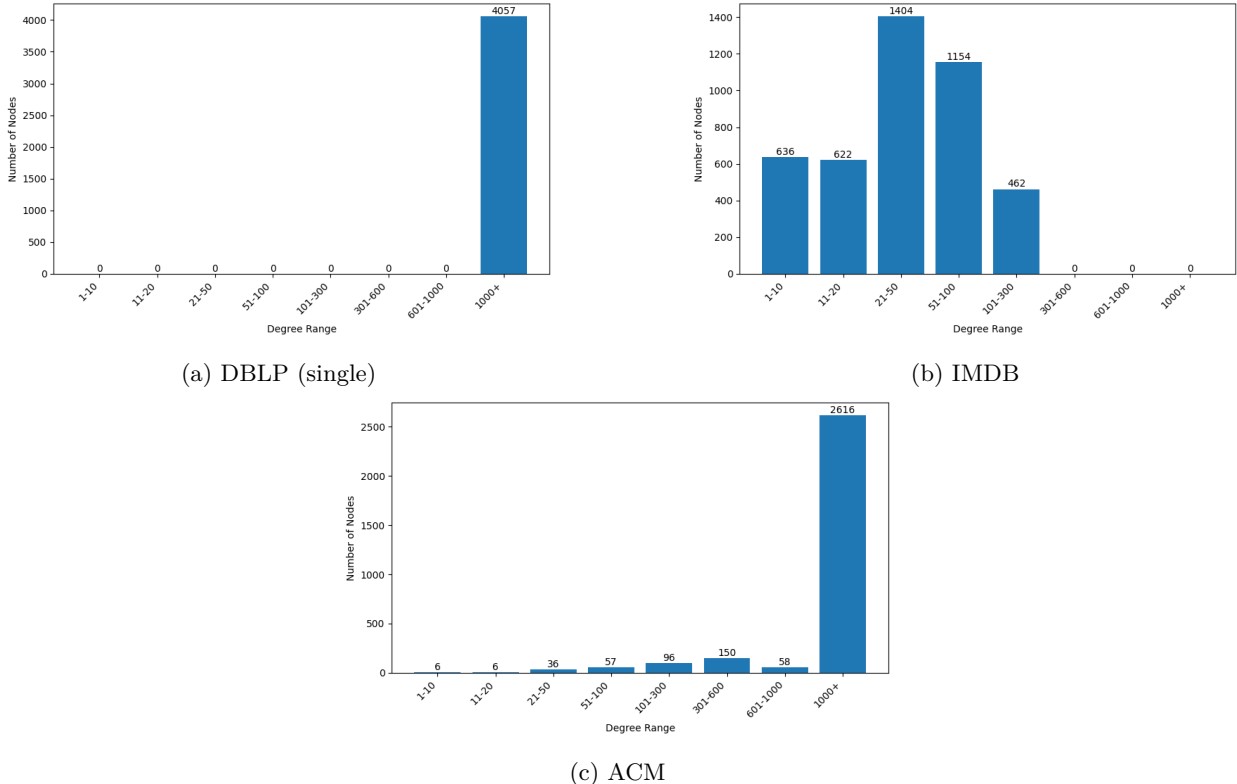

(a) DBLP (single)

(b) IMDB

(c) ACM

Figure 6: Node degree distributions for the computational graph (only involving meta-paths) of heterogeneous datasets.

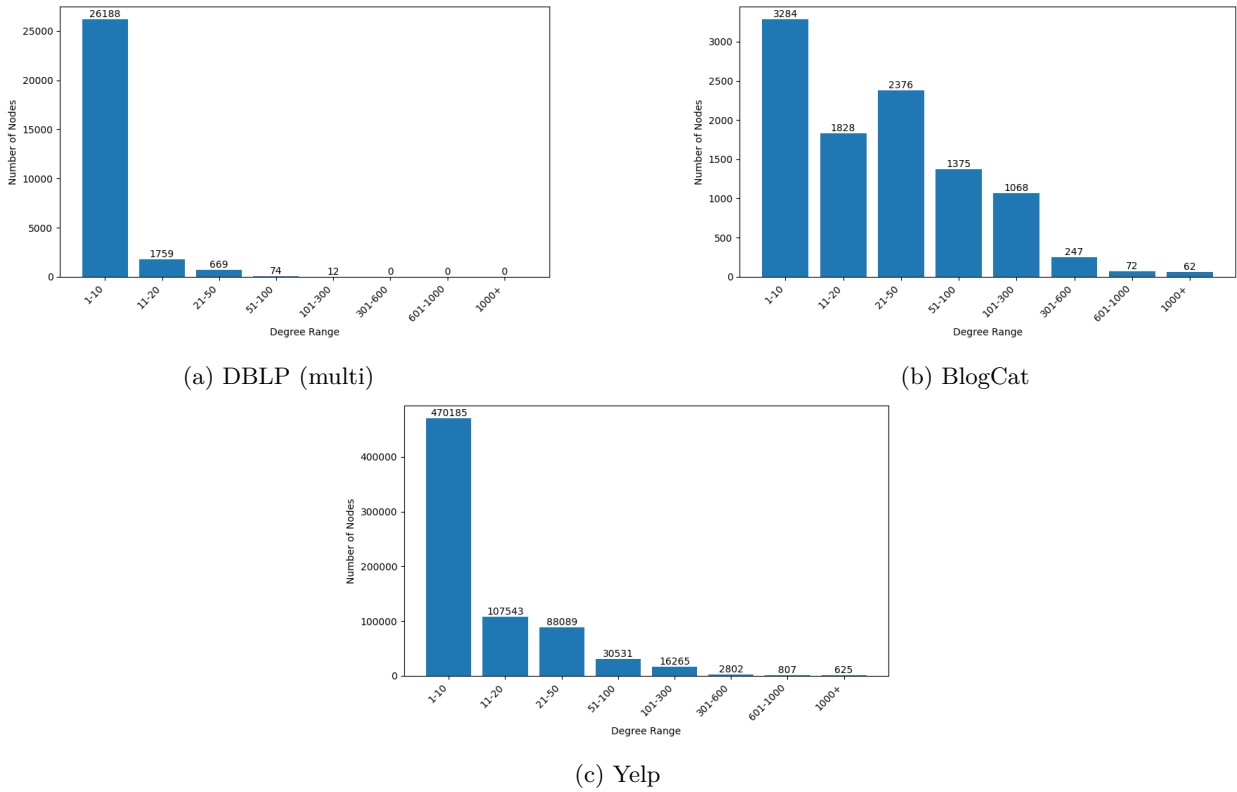

Figure 7: Node degree distributions for the multi-labeled datasets.

| | GraphSAGE | | | | GAT | | | |
|---|---|---|---|---|---|---|---|---|
| | NC | LP | $<$ | p | NC | LP | $<$ | p |
| CiteSeer | **96.24 ± 0.97** | 94.3 ± 1.32 | 10 | 0.002 | 90.26 ± 1.26 | **94.09 ± 0.93** | 100 | 0 |
| OGBN-Arxiv | 67.99 ± 3.76 | **70.95 ± 1.00** | 80 | 0.155 | **69.57 ± 0.17** | 65.69 ± 0.66 | 0 | 0 |
| Roman Empire | **80.27 ± 1.36** | 72.44 ± 1.11 | 0 | 0 | **42.41 ± 2.90** | 38.05 ± 2.52 | 10 | 0.002 |

Table 12: Test scores on representative datasets using models with three layers. Hidden channel dimensions are smooth interpolations of the input and output dimensions. p represents the p-value, we report 0 whenever it is smaller than 0.001.

original paper. The added nodes and message-passing edges will still yield added information in the LP representation.

## A.5 Hyperparameter analysis on embedding dimensions

Table 12 shows the test scores on selected datasets when the hidden channel embedding dimension are not fixed to 64, but instead smoothly interpolate between input and output embedding sizes. For example, if input feature has dimension 4000, output embedding has dimension 1000, then the three-layer models have embeddings 4000-3000-2000-1000.

## A.6 Time complexity analysis

To better understand the trade-off between the two representations, a high-level time complexity analysis is performed for the scenario of running GraphSAGE on a single-labeled homogeneous graph. Following the

description in Section 3.1, the two representations:

$$G_{NC} = (V, A, T, R) \quad \text{and} \quad G_{LP} = (V \cup \mathcal{L}, A \cup \{*\}, T \cup T', R \cup \{*'\}),$$

are studied. $G_{NC}$ has $|V|$ nodes and $|T|$ edges, where each node has a feature size of $d$. As a result of the introduced label nodes and edges, $G_{LP}$ will consist of $|V \cup \mathcal{L}|$ nodes and $|T \cup T'|$ edges, where each introduced label node has a feature vector of dimension $|\mathcal{L}|$.

Since GraphSAGE is the model of choice, each round of message passing will include an aggregation from a fixed number of neighbors. Let the maximum number of sampled neighbours be $S$. The forward pass of one GraphSAGE layer $l$, with input and output embedding sizes $\text{emb}_{l-1}$ and $\text{emb}_{l-1}$, has three distinct operations:

1. **Neighbourhood aggregation:** Every node aggregates embeddings from $S$ neighbours, each embedding having dimension $\text{emb}_{l-1}$. Over all $|V|$ nodes, this results in a time complexity of $\mathcal{O}(|V| \cdot S \cdot \text{emb}_{l-1})$.

2. **Message passing:** After aggregation, each node's current embedding is concatenated with the embedding of the neighbourhood. This concatenation step is $\mathcal{O}(\text{emb}_{l-1})$, followed by a multiplication by a weight matrix $\mathbf{W}^{\text{emb}_l \times 2\text{emb}_{l-1}}$ of time complexity $\mathcal{O}(\text{emb}_l \cdot 2\text{emb}_{l-1}) = \mathcal{O}(\text{emb}_l \cdot \text{emb}_{l-1})$. For all nodes, the time complexity of one round of message passing would be $\mathcal{O}(|V| \cdot \text{emb}_l \cdot \text{emb}_{l-1})$.

3. **Non-linear activation:** Activation functions are applied element-wise on $\text{emb}_l$-sized embeddings. This has a time complexity of $\mathcal{O}(|V| \cdot \text{emb}_l)$

The above operations give a total time complexity of

$$\mathcal{O}(|V| \cdot S \cdot \text{emb}_{l-1}) + \mathcal{O}(|V| \cdot \text{emb}_l \cdot \text{emb}_{l-1}) + \mathcal{O}(|V| \cdot \text{emb}_l) = \mathcal{O}(|V| \cdot \text{emb}_l \cdot \text{emb}_{l-1}).$$

The complexity of a backward pass is analogous to the forward pass due to reverse-mode automatic differentiation, which is used in PyTorch(Blakely et al., 2019). This gives a forward/backward pass time complexity of $\mathcal{O}(|V| \cdot \text{emb}_l \cdot \text{emb}_{l-1})$ for $G_{NC}$.

However, because of the heterogeneity of $G_{LP}$, the model architecture must take node and edge types into consideration (see Figure 4). This results in the architecture of the model for $G_{LP}$ having three times the amount of GraphSAGE layers. The time complexity of a forward/backward pass for the 3-layer model can be denoted as $T_{NC}$ for $G_{NC}$:

$$T_{NC} = \mathcal{O}(|V| \cdot \text{emb}_{L1} \cdot d) + \mathcal{O}(|V| \cdot \text{emb}_{L2} \cdot \text{emb}_{L1}) + \mathcal{O}(|V| \cdot \text{emb}_{L3} \cdot \text{emb}_{L2}) \tag{6}$$

and $T_{LP}$ for $G_{LP}$:

$$\begin{aligned} T_{LP} = {} & 2\mathcal{O}(|V| \cdot \text{emb}_{L1} \cdot d) + \mathcal{O}(|\mathcal{L}|^2 \cdot \text{emb}_{L1}) \\ & + 2\mathcal{O}(|V| \cdot \text{emb}_{L2} \cdot \text{emb}_{L1}) + \mathcal{O}(|\mathcal{L}| \cdot \text{emb}_{L2} \cdot \text{emb}_{L1}) \\ & + 2\mathcal{O}(|V| \cdot \text{emb}_{L3} \cdot \text{emb}_{L2}) + \mathcal{O}(|\mathcal{L}| \cdot \text{emb}_{L3} \cdot \text{emb}_{L2}). \end{aligned} \tag{7}$$

Equations 6 and 7 yield

$$T_{LP} = 2 \cdot T_{NC} + \mathcal{O}(|\mathcal{L}|^2 \cdot \text{emb}_{L1}) + \mathcal{O}(|\mathcal{L}| \cdot \text{emb}_{L2} \cdot \text{emb}_{L1}) + \mathcal{O}(|\mathcal{L}| \cdot \text{emb}_{L3} \cdot \text{emb}_{L2}).$$

Assuming $|V| > |\mathcal{L}|$ and $d > |\mathcal{L}|$, which hold for all datasets used as well as in general, we have

$$T_{NC} > \mathcal{O}(|\mathcal{L}|^2 \cdot \text{emb}_{L1}) + \mathcal{O}(|\mathcal{L}| \cdot \text{emb}_{L2} \cdot \text{emb}_{L1}) + \mathcal{O}(|\mathcal{L}| \cdot \text{emb}_{L3} \cdot \text{emb}_{L2}).$$

Finally, this yields

$$2 \cdot T_{NC} < T_{LP} < 3 \cdot T_{NC}. \tag{8}$$

### A.7    Number of learnable parameters

Building on Section A.6, a similar analysis can be made for the number of learnable parameters for the two representations. As previously stated, for each GraphSAGE layer, the weight matrix is $\mathbf{W}^{\text{emb}_l \times 2\text{emb}_{l-1}}$. Including the bias, there are $2 \cdot \text{emb}_l \cdot \text{emb}_{l-1} + \text{emb}_l$ learnable parameters per layer. Using the same scenario as in Section A.6, the number of learnable parameters $C_{NC}$ for the model used on $G_{NC}$ is

$$C_{NC} = \text{emb}_{L1} \cdot d + \text{emb}_{L2} \cdot \text{emb}_{L1} + \text{emb}_{L3} \cdot \text{emb}_{L2} + \text{emb}_{L1} + \text{emb}_{L2} + \text{emb}_{L3}. \tag{9}$$

Similarly, with the same the number of learnable parameters for the model used on $G_{LP}$ are

$$C_{LP} = 2 \cdot (\text{emb}_{L1} \cdot d + \text{emb}_{L2} \cdot \text{emb}_{L1} + \text{emb}_{L3} \cdot \text{emb}_{L2} + \text{emb}_{L1} + \text{emb}_{L2} + \text{emb}_{L3})$$
$$+ \text{emb}_{L1} \cdot |\mathcal{L}| + \text{emb}_{L2} \cdot \text{emb}_{L1} + \text{emb}_{L3} \cdot \text{emb}_{L2} + \text{emb}_{L1} + \text{emb}_{L2} + \text{emb}_{L3}. \tag{10}$$

With the assumption of $d > |\mathcal{L}|$, Equations 9 and 10 yield

$$C_{NC} > \text{emb}_{L1} \cdot |\mathcal{L}| + \text{emb}_{L2} \cdot \text{emb}_{L1} + \text{emb}_{L3} \cdot \text{emb}_{L2} + \text{emb}_{L1} + \text{emb}_{L2} + \text{emb}_{L3},$$

which gives

$$2 \cdot C_{NC} < C_{LP} < 3 \cdot C_{NC}. \tag{11}$$

## B    GCN

We conduct experiments for GCN on various datasets. For each dataset, running GCN on representation $G_{NC}$ is standard. However, applying GCN on $G_{LP}$ requires additional modifications. The GCN convolution is defined as

$$H^{k+1} = \sigma\left(\tilde{D}^{-\frac{1}{2}} \tilde{A} \tilde{D}^{-\frac{1}{2}} H^k W^k\right), \tag{12}$$

where $W^k$ is the k-layer weights and $\tilde{A} = A + I_N$ is the adjacency matrix with added self-loops. Diagonal matrix $\tilde{D}_{ii} = \Sigma_j \tilde{A}_{ij}$.

In the heterogeneous $G_{LP}$ graph two issues arise. Firstly, Equation 12 is defined only for square adjacency matrices and therefore cannot handle the bipartite relations (T' in Definition 3.2). Secondly, it assumes all nodes have the same embedding dimension and are initialised similarly, which is not the case for sets of nodes $V$ and $\mathcal{L}$ in $G_{LP}$.

To solve this, we decompose the computation graph in a HeteroConv similar to Figure 4. Taking an example of a paper-citation graph (with label nodes in $G_{LP}$) the HeteroConv is:

$$\text{HeteroConv}\Big\{\underbrace{(\text{p}, \text{cites}, \text{p})}_{\text{GCNConv}}, \underbrace{(\text{p}, \text{is}, \text{l})}_{\text{GraphConv}}, \underbrace{(\text{l}, \text{rev\_is}, \text{p})}_{\text{GraphConv}}\Big\}_{\text{aggr=mean}},$$

where separate channels are mean-aggregated at the node level after each layer. GCN is preserved on the homogeneous citation subgraph, while the bipartite edges between papers and labels are handled by a simple GraphConv(Morris et al., 2019) without edge weights, defined as:

$$\mathbf{h}_i^{k+1} = \mathbf{W}_1^k \mathbf{h}_i^k + \mathbf{W}_2^k \sum_{j \in \mathcal{N}(i)} \mathbf{h}_j^k, \tag{13}$$

which allows different embedding dimensions for source ($x_i$) and target ($x_j$) nodes in message passing.

Results for the heterophilic/homophilic and multi-labelled datasets are shown in Table 13. It can be observed that for some datasets such as CiteSeer, models perform similarly on the two representations. For most other datasets, the LP representation either yields significantly worse performance, or sometimes wildly different results that varies across seeds (such as for Roman Empire). This is likely due to the ad hoc definition of our heterogeneous GCN.

| | GCN | | | |
|---|---|---|---|---|
| | NC | LP | $<$ | p |
| CiteSeer | $\mathbf{95.48 \pm 0.92}$ | $95.41 \pm 1.08$ | 40 | 0.876 |
| Cora-ML | $\mathbf{86.80 \pm 1.19}$ | $62.33 \pm 16.9$ | 0 | 0.001 |
| OGBN-Arxiv | $\mathbf{70.86 \pm 0.00}$ | $51.24 \pm 23.9$ | 10 | 0.140 |
| Chameleon | $\mathbf{67.23 \pm 3.98}$ | $44.47 \pm 6.23$ | 0 | 0 |
| Squirrel | $\mathbf{39.15 \pm 2.81}$ | $31.92 \pm 3.53$ | 10 | 0.008 |
| Roman Empire | $\mathbf{41.55 \pm 0.75}$ | $38.42 \pm 12.3$ | 40 | 0.443 |
| DBLP(multi) | $\mathbf{94.29 \pm 0.41}$ | $56.76 \pm 0.13$ | 0 | 0 |
| BlogCat | $\mathbf{80.04 \pm 0.96}$ | $4.86 \pm 1.42$ | 0 | 0 |
| Yelp | $\mathbf{42.41 \pm 0.32}$ | $16.0 \pm 0.47$ | 0 | 0.01 |

Table 13: Test scores using GCN with 3 layers. $<$ represents the percentage of runs where $G_{LP}$ is a better representation than $G_{NC}$. p represents the p-value, we report 0 whenever it is smaller than 0.001.

## C  GIN

GIN was initially proposed as an 1-WL expressive architecture to solve graph classification problems involving non-isomorphic graphs that may be difficult to separate with other architectures. Although there is no evidence to suggest higher expressivity (when measured by WL) helps with labelling nodes on our sets of graphs, we conduct experiments of GIN on the 6 heterophilic/homophilic datasets.

A GIN layer is defined as:

$$h_i^{k+1} = \text{MLP}^k\big((1+\varepsilon)h_i^k + \sum_{j \in \mathcal{N}(i)} h_j^k\big). \tag{14}$$

While applying GIN on $G_{NC}$ is again standard, directly adding $h_i$ and summed $h_j$ implies it does not work on $G_{LP}$.

Therefore, for each node type, we add a learnable linear layer that maps the input node embeddings into a common latent space with the same dimension, and then use the GINConv layer **to_hetero**$\big(\text{GINConv}(...)\big)$ in the same way as the GraphSAGE and GAT (Figure 4) for $G_{LP}$.

Our results are shown in Table 14. GIN on $G_{LP}$ sometimes produces wildly different results than on $G_{NC}$. It performs generally better on $G_{NC}$ and sometimes indifferent (e.g Squirrel).

| | GIN | | | |
|---|---|---|---|---|
| | NC | LP | $<$ | p |
| CiteSeer | $\mathbf{95.27 \pm 1.31}$ | $93.88 \pm 2.98$ | 30 | 0.1994 |
| Cora-ML | $\mathbf{86.43 \pm 1.69}$ | $81.73 \pm 3.05$ | 0 | 0 |
| OGBN-Arxiv | $\mathbf{68.36 \pm 0.57}$ | $60.94 \pm 3.20$ | 0 | 0.006 |
| Chameleon | $\mathbf{43.33 \pm 8.68}$ | $29.96 \pm 6.54$ | 10 | 0.001 |
| Squirrel | $32.50 \pm 2.96$ | $\mathbf{33.88 \pm 3.22}$ | 20 | 0.499 |
| Roman Empire | $\mathbf{45.08 \pm 1.99}$ | $37.82 \pm 6.12$ | 10 | 0.004 |

Table 14: Test scores using GIN with 3 layers. $<$ represents the percentage of runs where $G_{LP}$ is a better representation than $G_{NC}$. p represents the p-value, we report 0 whenever it is smaller than 0.001.

