# OpenReview forum: "Formulating Node Labelling as Node Classification or Link Prediction in Different Graph Representations"
_TMLR — Accepted by TMLR_

### Review · Reviewer_N3cy · 2025-05-20

**Summary Of Contributions:**

This work compares two different problem settings that achieve the same goal. Namely, it compares node classification and link prediction on a modified graph instance. The authors discuss strategies for fair comparison and finally conduct experiments on 12 datasets observing different graph properties correlating with performance.

**Audience:**

Yes

**Broader Impact Concerns:**

There are no such concerns.

**Claims And Evidence:**

No

**Requested Changes:**

1. Page 1 "users tagging music with genre". Why are you using quotes here? What are you quoting? Also, the quote symbols used are not proper. This example is generally poorly chosen since this work focuses on the comparison between node classification and link prediction, but this example compares labeled link prediction to hyperedge prediction.
2. Page 3 partial labelling function L. Why are you referring to this function as a partial labeling function? It assigns every vertex a set (which might be empty) of labels. It is not clear whether you are implying that there are labels missing or simply referring to the fact that some nodes have an empty set of labels.
3. Page 3 \mathcal V_L. Due to the previous issue, it is unclear how \mathcal V_L is defined.
4. Page 3 \beta_K definition. In this definition, you use \mathcal N^K, which has not been defined. Likely you are referring to the k-hop neighborhood, however, there are two common definitions of this, one including all nodes with paths of length at most K, the other including all nodes with paths of length exactly K.
5. Page 3 message passing GNN. You seem to be using the definition of the Message Passing Neural Network (not Graph Neural Network) defined by Gilmer ("Neural Message Passing for Quantum Chemistry" 2017) without citing this work. Please correct this.
6. Page 3 Formally the message passing step as (should be is)
7. Page 3, Equation 1. There is a missing closing brace.
8. Page 3 after equation 1. You state that the UPDATE function is a learnable MLP, however, the UPDATE functions takes as input a tuple, which is not a standard input for an MLP. This should be clarified.
9. Page 3 Model choice. You chose to evaluate GraphSAGE and GAT but do not justify your choice. Standard alternatives that are more common in general are the graph convolutional network (GCN) and graph isomorphism network (GIN). Please justify this choice adequately based on related work.
10. Page 3 The GNN formula ... . Please specify what you are referring to with "GNN formula".
11. Page 3 \mathcal A \times \mathcal R \times \mathcal A. It is unclear from this description how you would use equation 1 to deal with heterogenous graphs.
12. Page 4 abstract graph. What is an abstract graph? This has not been defined.
13. Page 4 use mostly default font ... . This description is not at all adequate. Please specify in detail how you transform the original graph using calligraphic font into the graphs in definitions 3.1-3.4.
14. Page 4 Definition 3.3-3.4 You use P_{NC} and P_{LP}, however, it is not clear that these are descriptions of these definitions. Perhaps add more surrounding text explaining the context and add the description P_{NC} and P_{LP} into their definition defining what exactly this term refers to.
15. Page 4 Definition 3.4 (V \times *' \times V). There is a set missing here. It should be (V \times \{*'\} \times V).
16. Page 4 and beyond set unions. You are commonly defining set unions that are disjoint. Perhaps you could use \cupdot to clarify that these are disjoint unions and thus partitions of some larger set.
17. Page 4 ", both as information used by the learning method". I do not understand the implication or point of this part of the sentence.
18. Page 4 equation 2-5. The description of these equations and, thus, your transformation is inadequate. You should clearly define all subindices like train_mp and train_sup. You should also clearly define what you mean by message passing and supervision edges. Further, you should give a clear motivation for this transformation and describe your goal.
19. Page 5 Figure 2. Figure 2 is not sufficiently explained. What are dotted and non-dotted lines referring to? How are these graphs equivalent? You should extend the description of Figure 2 to fully explain what is depicted.
20. Page 5 Section 4.1. You write |T'_{train_mp}\setminus T'_{train_sup}| = 3/7; however, these two sets are likely disjoint, and the quantity you are computing is unclear. It would be better to state that the size of the first set is 30% of all edges and that the size of the second set is 70% of edges.
21. Starting on Page 6 Section 4.3. You never describe how hyperparameter tuning was performed. Importantly, you are comparing models that might need different parameters to perform well. If no hyperparameter tuning is performed then how do you pose that any of your evaluations are comparable?
22. Starting in Table 1. In your tables, you mark the best-performing model in bold. Importantly, this should be described in each table's caption. However, you do not consider the statistical significance of the difference. For example, in Table 1 Cora-ML using GraphSAGE NC vs LP, the difference is within one standard deviation and thus likely not significant. Please perform a t-test significance test using your 10 (or in some cases less) seeds and indicate in each table whether the difference is significant. For example, if it is not significant, you could mark both as bold or the second-best performing one (which is only insignificantly worse) with an underline. After this, you should reevaluate your conclusions. I suspect that you will not be able to draw many conclusions, which would invalidate some of your points.
23. Page 7 and 9. It seems that there are some formatting issues in your manuscript. Pages 7 and 9 contain empty spaces and are not formatted well. Please resolve this issue and place the respective tables where they are used.
24. Page 8 mega-nodes. I am not familiar with this term.
25. Page 8 SBM graphs. In the description in Section 4.6, you consider SBMs built with 4 blocks, but you later refer to "We assign binary labels to each of the two blocks of nodes". Are there two blocks or 4 blocks?
26. Page 8 random flipping. It is not clear how randomly flipping node labels makes the graphs non-trivial.
27. Page 9 Table 6. Table 6 is too big for the page.
28. Page 6 Figure 3. It is bad style to include images with text in them where this can be avoided. Remove the text from both images and replace this with subcaptions.
29. Page 10 Related Work. Your related work on Rewiring is rather extensive, even though it is unclear how rewiring is related to your work. On the other hand, related work closely related to this manuscript is rather sparse. Consider doing an extended literature review. For example, you did not include Fadaee and Haeri ("Classification using Link prediction" 2019) or Bligic et al. ("Combining Collective Classification and Link Prediction" 2007) which a cursory glance revealed suggesting there are more works that are closely related to your work and perhaps implying that this is not the first time this question has been considered. Consider further narrowing of your claims in the Introduction regarding your work.
30. Page 10 Section 6 Title. Discussion and Future Work is filled with conclusion (not discussion), future work, and limitations (not mentioned). Perhaps consider separating these into individual sections, or choose a better section title.
31. General. In general, I suggest that you expand the captions on all figures and tables. Include a take-home message or conclusion as a first sentence for each followed by a detailed description that fully summarizes the figure or table. Generally, I also suggest that you move tables either to the top or bottom of each page for legibility (consider Figure 1 which is surrounded by text).

**Strengths And Weaknesses:**

# Strengths
- The issue of problem encoding is an interesting topic that has not been widely studied.
- The conclusions, if valid, could impact future model choice and design.
- The evaluations are fairly broad, even though they are limited to single graph datasets.

# Weaknesses
- This manuscript is not well-polished.
- There are a number of lacking descriptions leading to challenges in comprehension.
- There are numerous typos and similar mistakes leading to challenges in comprehension.
- The problem setup is not well-defined, which could imply a vacuous conclusion.
- Conclusions are drawn without the use of statistical significance, which could imply a vacuous conclusion.

---

> ### Author Response · Authors · 2025-06-28
>
> Thanks for the detailed review and suggestions. We have applied the suggestions regarding wordings, definitions and clarifications. We respond to several of the questions that we think benefit from additional clarifications in detail below.
>
> 9.GCN and GIN: We added them to the background section 2.1. They are indeed common alternatives. However, neither model works out-of-the-box for $G_{LP}$. GCN requires a homogeneous graph (square adjacency) and both models require equal initial node embedding dimensions. We added in Section 4.6 (final paragraph) and Appendix B and C why and how we extended GCN and GIN to $G_{LP}$. We added experiments of the two models on the relevant datasets.
>
> 12-14. We clarified the definitions in Section 3.1. We wanted to stress the approach we take when studying the representation. Firstly, there is some given data and end problem to solve, such as assigning papers with subject categories for citation network. There are various concrete representations of the citation network, such as $G_{NC}$, and $G_{LP}$. As a result, there are possibly different ways of defining the end problem as learning tasks on graph. The special phenomenon of $G_{NC}$ and $G_{LP}$ is that the end problem turns out to be naturally different. We have now reworded to make this clearer, so that the readers can understand the approach and intuition behind the work.
>
> 21.This is a great question and we have also tried to make sure the experiments are fair and results are comparable, since it is uncommon to compare across tasks and representations (and models). We have in various places specified how they are determined. For example, splits taken from prior work when available, otherwise random split with the 80-10-10 ratio. The mapping of a split for NC (all datasets come in the $G_{NC}$ representation from prior benchmarks) to a split for LP follows exactly Section 3.2. For the key hyperparameters, such as num\_layers, we report results for both 2 and 3 layers which are common for the given datasets. We also included some analysis using different embedding dimensions and message/supervision ratio.
>
> 22.This is a great suggestion. We added p-value column in all tables in both the main text and the appendices. We have modified our discussions and conclusions in text whenever appropriate. For example, we now point out for GAT on ogbn-arxiv\_{LP} and ogbn-arxiv_{NC} is insignificant (Section 4.3), and removed a claim about DBLP(single)(Section 4.4). Most of the differences remain significant and other claims are still valid. Our column < (NC < LP) compares scores under the same seed (which affects the random splits), which offers another statistical perspective of how often one representation is better than the other when only comparing on the same splits.
>
> 25-26.We clarified how we assign the node labels. While the graph generation and node labelling process is random, the resulting labelling function that the models need to learn is non-trivial. The added node flipping simply introduces additional noise, which the models need to learn to not overfit on.
>
> 29.These are indeed two important references that we overlooked. We have added “Fadaee \& Haeri, Classification using Link prediction” to the related work. Their proposed classical algorithm that assigns labels to nodes in a graph using a variant of Adamic-Adar, with the graph created using kNN, is indeed very relevant to our setting. However, they did not study the problem from the point of view of understanding how graph representations (and tasks) impact message-passing GNNs in graph learning. Our work becomes an extension of theirs. We have also added “Bligic et al., Combining Collective Classification and Link Prediction” to the related work. This work is similar to the other references we had that studies node classification and link prediction over graphs with missing labels and links jointly, which is a different problem than ours.
>
> All changes are highlighted in blue in the revised version.

---

> > ### Comment · Reviewer_N3cy · 2025-07-14
> > **Slight Correction**
> >
> > Thank you for your response. I am satisfied with most of your changes. The most significant remaining error is your definition of L:\mathcal V \to \mathcal P(\mathcal L) \setminus \emptyset. The issue is that, as written, L is defined on all nodes in \mathcal V. However, you state that it should only be defined on \mathcal V_L. To correct this issue, you should just change the definition to L:\mathcal V_L \to \mathcal P(\mathcal L) \setminus \emptyset, i.e., change \mathcal V to \mathcal V_L, then L is only defined on this set. However, you should then reorganize your sentences slightly, defining \mathcal V_L before you define L, perhaps also giving a name to all unlabeled nodes. But I leave this to your discretion.

---

> > > ### Author Response · Authors · 2025-07-16
> > >
> > > Thanks for going through our revision and the acknowledgement. If we define L as a partial function, it is well-defined in its current form. If L is a function, then we need to define the domain $\mathcal{V}_L$ upfront. The debate becomes which formulation is easier and clearer to work with in subsequent definitions and makes the NC/LP comparison more understandable.
> > >
> > > We will look at defining $\mathcal{V}_L$ and a function $L$ on domain $\mathcal{V}_L$, and reorganize related definitions. We will apply the changes in the final version if it makes the paper presentation better. Thanks for the suggestion.

---

### Review · Reviewer_U6bd · 2025-05-29

**Summary Of Contributions:**

The study examines how different graph encodings influence node-label prediction with message-passing GNNs, demonstrating that a link-prediction formulation can, in certain settings, surpass the conventional node-classification approach.

**Audience:**

Yes

**Claims And Evidence:**

Yes

**Requested Changes:**

I wonder how does the different graph representations affect non-parametric graph learning methods such as label propagation. It would be interesting to explore in this perspective.

**Strengths And Weaknesses:**

Strengths:

- This work introduces a novel, data-centric perspective on graph representation design.
- The paper conducts comprehensive experiments to elucidate how alternative graph encodings influence model performance.

Weaknesses:

- No theoretical insights explaining why LP helps in sparse graphs; a more in-depth analysis could strengthen the paper.
- Only vanilla GraphSAGE/GAT/HAN are evaluated; more powerful architectures might behave differently.

---

> ### Author Response · Authors · 2025-06-28
>
> Thanks for acknowledging our contributions and for the suggestions. We added the label propagation algorithm as a non-parametric model to Section 4.7 (Page 10). However, note that LPA fundamentally solves the node classification task, even on $G_{LP}$. It therefore provides intermediate results instead of a direct comparison between $P_{NC}$ and $P_{LP}$.
>
> > Only vanilla GraphSAGE/GAT/HAN are evaluated; more powerful architectures might behave differently.
>
> In addition to LPA, we also added GCN and GIN in Section 4.7 and Appendix B, C.
>
> > No theoretical insights explaining why LP helps in sparse graphs; a more in-depth analysis could strengthen the paper.
>
> This is true. We refrain from making conclusive theoretical results about when $G_{LP}$ is a better representation than $G_{NC}$. Our comprehensive experiments suggest empirical trends of some models performing better on $G_{LP}$ for a certain distribution of graphs (Section 4.6, Page 8) validated with additional random graphs drawn from the distribution. A well-defined and useful formal result is very non-trivial, since it will at least be conditioned on a variety of terms, such as sparsity, homophily index and fine-grained degree distribution, as well as model-related parameters such as message/aggregation functions. We believe it will require more than one additional work to reach such conclusion.
>
> All changes are highlighted in blue in the revised version.

---

### Review · Reviewer_EsZR · 2025-06-24

**Summary Of Contributions:**

This paper examines how the same graph modelling problem can be formulated either as node classification (NC) or link prediction (LP), depending on how the underlying graph is constructed. It conduct a through empirical study across different datasets and settings, based on two graph backbone models (GraphSAGE and GAT). The core conclusion is that the choice of graph structural representation (how it is constructed) can significantly impact model performance.

**Audience:**

Yes

**Broader Impact Concerns:**

N/ A

**Claims And Evidence:**

No

**Requested Changes:**

Please refers to the weakness section, I think adding more theoretical analysis is not required but would be nice if the authors could address it somehow (W2). I would expect the authors to response to all my other points in the weakness section (W1, W3-W5).

**Strengths And Weaknesses:**

**strengths**

1. The paper explains a novel empirical problem and important for graph representation learning community but often neglect in practise
2. A detailed empirical study is conducted with careful handle of data split and evaluation metrics
3. The formulation of transformation from NC to LP tasks (seems) is novel
4. The paper is generally easy to follow and well-written

**weakness**

1. All empirical experiments and results are not very useful (e.g. not much to say about when to formulate LP or NC task, what is the trade-offs, why sometimes one way of formulation is preferred and what should be done when this happens)
2. No theoretical analysis and hence make this paper sounds like a scientific report
3. Lack of reason why pick GraphSAGE and GAT and why under transductive setting only
4. I think the formulation of graph structure is a very strong inductive bias and should be data-dependent, not sure if I am convinced on results reported based on architeture.
5. May lack of variety of models used for evaluation, and the motivation for this research could framed a bit better

---

> ### Author Response · Authors · 2025-06-28
>
> Thanks for acknowledging our contributions and for the suggestions. We comment on each of the reviewer's concern below.
>
> > All empirical experiments and results are not very useful (e.g. not much to say about when to formulate LP or NC task, what is the trade-offs, why sometimes one way of formulation is preferred and what should be done when this happens)
>
> We need to first understand whether different representations of graphs makes a difference, which is important to know, intuitively clear but rarely studied - echoed by the reviewer. Among all the possible general representations, we identified two where the formal tasks even differs. Through our definitions and comprehensive experiments the empirical conclusion is positive. We take a step to draw heuristics about when LP is better and validate that with random graphs (Section 4.6, Page 8). We acknowledge this is only an initial step towards a general criterion. Regarding the trade-offs between the two tasks, we agree that this is a valid and important point. In response, we have added analyses of how the time complexity and number of learnable parameters differ between the two tasks (Appendix A.6, A.7).
>
> > No theoretical analysis and hence make this paper sounds like a scientific report.
>
> Following the previous point, we acknowledge that in order to provide a general criterion about when to use NC/LP,  theoretical proofs are required. This could be even further extended into a general recipe of defining the optimal representation (beyond $G_{NC}$, $G_{LP}$). Our results already suggest that such as receipt is very non-trivial, since it is likely to be dependent on a variety of graph properties (sparisity, homophily index, fine-grained degree distribution) and model hyperparameters (message/aggregate functions). We do not make any general claim theoretically and we believe this alone requires a stream of future work.
>
> > Lack of reason why pick GraphSAGE and GAT and why under transductive setting only.
>
> This is a very valid question. We added experiments for GCN and GIN (Section 4.7, Appendix B, C) as well as label propagation (as a classical non-parameteric algorithm for intermediate insights, Section 4.7). We stated in Section 7 (Page 12) that extending the work to inductive setting is an important future work and highlights the challenges. Similar empirical conclusions for the inductive setting will require a large set of experiments similar to ours.
>
> > I think the formulation of graph structure is a very strong inductive bias and should be data-dependent, not sure if I am convinced on results reported based on architeture.
>
> This is true. Our results are data-dependent. For example, GS performs better on ogbn-arxi$v_{LP}$ than ogbn-arxi$v_{NC}$, but worse on CiteSee$r_{LP}$ than CiteSee$r_{NC}$. It is additionally model-dependent. This highlights the challenge towards W2.
>
> > May lack of variety of models used for evaluation, and the motivation for this research could framed a bit better.
>
> Additional models are now added to address W3. We also added clarifications to some of our definition as per other reviewers' suggestions. Together with the introduction (highlighted by Figure 1) and the research questions (in bullet points, Page 2), we believe the motivations should now be clear.
>
> All changes are highlighted in blue in the revised version.

---

### Decision · Action_Editor_Sjq6 · 2025-07-23

**Recommendation:** Accept with minor revision

**Audience:**

Yes

**Audience Explanation:**

Most reviewers agree that the paper and its results are interesting, but need to be better positioned and compared with other methods and techniques. I strongly encourage the authors to address these points in the final version of the paper.

**Claims And Evidence:**

Yes

**Claims Explanation:**

The reviewers appreciated the impressive experimental sweep across graph representations and GNN variants, but many still struggled to distill clear, actionable heuristics from the results. Reviewers mostly agree that claims and evidence are accurate and that there is interest for this work within the TMLR's audience. Thus, it aligns with TMLR's policy. Nonetheless, the reviewers strongly suggest that that the authors (i) provide a non-parametric baseline (e.g., label propagation) to clarify when/why a representation helps; (ii) turn observed trends into explicit, statistically validated criteria (with significance tests/effect sizes) and (iii) provide theoretical or at least principled explanations. Please address these points in the camera-ready to better reflect the contribution of the paper.

---

> ### Author Response · Authors · 2025-07-30
>
> We thanks all the reviewers and the AE for all the suggestions throughout the review process. The camera ready version now contains the all the proposed changes, with (i) in Section 4.7, (ii) in all the tables and Section 4.6, (iii) in Section 4.6.